# Critical Roles of Polycomb Repressive Complexes in Transcription and Cancer

**DOI:** 10.3390/ijms23179574

**Published:** 2022-08-24

**Authors:** Guan-Jun Dong, Jia-Le Xu, Yu-Ruo Qi, Zi-Qiao Yuan, Wen Zhao

**Affiliations:** State Key Laboratory of Esophageal Cancer Prevention and Treatment, Key Laboratory of Advanced Pharmaceutical Technology, Ministry of Education of China, School of Pharmaceutical Sciences, Zhengzhou University, Zhengzhou 450001, China

**Keywords:** PRC1, PRC2, transcriptional regulation, cancer, inhibitors

## Abstract

Polycomp group (PcG) proteins are members of highly conserved multiprotein complexes, recognized as gene transcriptional repressors during development and shown to play a role in various physiological and pathological processes. PcG proteins consist of two Polycomb repressive complexes (PRCs) with different enzymatic activities: Polycomb repressive complexes 1 (PRC1), a ubiquitin ligase, and Polycomb repressive complexes 2 (PRC2), a histone methyltransferase. Traditionally, PRCs have been described to be associated with transcriptional repression of homeotic genes, as well as gene transcription activating effects. Particularly in cancer, PRCs have been found to misregulate gene expression, not only depending on the function of the whole PRCs, but also through their separate subunits. In this review, we focused especially on the recent findings in the transcriptional regulation of PRCs, the oncogenic and tumor-suppressive roles of PcG proteins, and the research progress of inhibitors targeting PRCs.

## 1. Introduction

Aberrant regulation of epigenetic pathways is thought to be a frequent event in cancer. Understanding the roles of these epigenetic regulators has facilitated the discovery of new therapeutic approaches for cancer [1]. Polycomb group (PcG) proteins are a group of widely studied epigenetic regulators, originally discovered in *Drosophila melanogaster* and found to be functionally and compositionally conserved in other animals, including *Caenorhabditis elegans*, mice, and humans [2,3,4]. The PcG proteins play a critical role in the progression of cancer forming multimeric complexes involved in transcriptional repression, including Polycomb repressive complex 1 (PRC1) and Polycomb repressive complex 2 (PRC2) [5].

Polycomb repressive complexes (PRCs) mainly mediate transcription and gene expression by regulating post-translational modifications of histones, in which PRC1 catalyzes the mono-ubiquitination of histone H2A at Lys119 (H2AK119ub1), whereas PRC2 catalyzes mono-, di-, and trimethylation of histone H3 at Lys27 (H3K27me1, H3K27me2, and H3K27me3) [6,7]. PRC2-catalyzed H3K27me3 is a hallmark of transcriptional silencing. Furthermore, PRC1 and PRC2 were found to spatially converge on the same sites in the genome to form Polycomb chromatin domains, with H2AK119ub1 and H3K27me3 uniquely enriched in these domains [8,9,10]. Gene repression is thought to be mediated by PRC1 and PRC2 cooperatively, although the specific mechanisms have not been fully defined. As shown in Figure 1, Polycomb complexes make up the catalytic core, and they bind accessory proteins to constitute distinct PRC1 and PRC2 complexes. In recent years, it has gradually emerged that PRC1 is not a single complex, and that there are at least eight different complexes [11]. These complexes can be further divided into canonical PRC1 (cPRC1) and non-canonical PRC1 (ncPRC1) depending on whether they contain one of two homologous proteins, YY1-associated factor 2 (YAF2) and RING1 and YY1 binding protein (RYBP), or a Chromobox (CBX) protein, respectively [12]. Moreover, proteomic and biochemical analyses have revealed that both cPRC1 and ncPRC1 possess E3 ubiquitin ligase RING1A/B, a core subunit that catalyzes the ubiquitination of histone H2A [12]. In mammals, PRC2 mainly contains four core subunits, enhancer of zeste homolog 1/2 (EZH1/2), embryonic ectoderm development (EED), suppressor of zeste 12 (SUZ12), and retinoblastoma protein-associated proteins 46/48 (RBAP46/48) [13,14]. In addition, these core proteins associate with different cofactors to create two distinct PRC2 variants, PRC2.1 and PRC2.2 (Figure 1B).

Studies have shown that the expression level of EZH2, the core catalytic subunit of PRC2, is positively correlated with tumor grade in prostate and breast cancers and PRC2 is initially thought to have an oncogenic function [15,16,17]. Subsequently, catalytically hyperactivating mutations of EZH2 in somatic cells were identified in patients with non-Hodgkin lymphoma (NHL), indicating that PRC2 is critically involved in the progression of lymphoma [18,19,20]. However, loss-of-function mutations in the catalytic subunit of PRC2 or loss of other components are found in some cancers (such as leukemia, myeloproliferative neoplasms, and malignant peripheral nerve sheath tumors), suggesting that PRC2 can also have tumor-suppressive functions [21,22,23]. Similar to PRC2, PRC1 also plays dual roles as an oncogenic and tumor suppressor. Its E3 ubiquitin ligase RING1A/B, subunits CBX2/4, and Polycomb group ring finger 1/4 (PCGF1/4) are oncogenic in leukemias, lymphomas, gliomas, and other tumors [24,25,26,27,28]. In contrast, some subunits of PRC1 have also been reported to have tumor-suppressive roles such as CBX6 and PCGF2 in breast cancer [29,30,31]. As discussed above, PRCs have both oncogenic and tumor-suppressive functions that may limit the use of inhibitors and highlight the importance of expounding the specific function of PRCs in the tumor.

In this review, we provide an overview of the diverse PRC1 and PRC2 variants generated by different combinations of PcG proteins. Additionally, we discuss the classical functions of PRCs as well as the recently identified novel functions of PcG proteins, providing some references for therapeutic directions of related inhibitors. Overall, we focus on two major themes: functions of PRCs in mammalian transcription, and the dual roles of core and accessory subunits of PRCs in cancer.

## 2. Composition of PRC1 and the Role It Plays in Transcription

PRC1 was first purified from *Drosophila melanogaster* embryos, which is mainly composed of Sex Combs Extra (Sce), Polycomb (Pc), Posterior Sex Combs (Psc), Polyhomeotic (Ph), and Sex Comb on Midleg (Scm) [32]. A similar complex containing the core subunits of *D. melanogaster* PRC1 and various homologous proteins was later purified in mammalian cells [4]. Proteomic and biochemical analyses over the last decade have revealed the enormous complexity of mammalian PRC1, which is now known to include many canonical and non-canonical complex variants (Figure 1A).

### 2.1. Composition of PRC1 and Its Ubiquitin Ligase Activity

All PRC1 variants contain the E3 ubiquitin ligase RING1A/B and are designated PRC1.1-PRC1.6 based on the six PCGF proteins they contain [12]. They are generally categorized as canonical PRC1 (cPRC1) or non-canonical PRC1 (ncPRC1, Figure 1A), a nomenclature that reflects the compositional similarity of these complexes to PRC1 in *D. melanogaster*. The mammalian cPRC1 complexes assemble around either PCGF2 or PCGF 4 (also referred to as MEL-18 or BMI-1) and include five chromodomain proteins (CBX2/4/6/7/8), three Polyhomeotic subunits (PHC1/2/3) and SCM homologs (SCMH1, SCML1/2), which is largely consistent with the composition of the originally discovered *Drosophila* PRC1 [4,12,33,34,35]. Six PCGF proteins were present in ncPRC1 variants, but only PCGF2 and PCGF4 were found in the cPRC1 [12]. Thus, PRC1.2 and PRC1.4 can be further subdivided into cPRC1.2/4 and ncPRC1.2/4. In addition, the ncPRC1 complexes were also composed of RYBP or its homologous protein YAF2 and various PCGF-specific cofactors [12,33]. Furthermore, a major difference between cPRC1 and ncPRC1 variants is whether their occupancy on chromatin is dependent on PRC2-deposited H3K27me3, and the CBX proteins of cPRC1 can bind to H3K27me3, which is required for their occupancy on chromatin [33]. By contrast, the RYBP or YAF2 subunits of ncPRC1 variants can bind chromatin independently of PRC2 proteins and their activity significantly stimulates the E3 ubiquitin ligase activity of RING1B in vitro [8,33,34,36]. An in-depth discussion of the specific classification of PRC1 is detailed in REF 5.

A key function of the mammalian PRC1 is to catalyze H2AK119ub1, an epigenetic modification closely associated with PRCs-mediated gene silencing [7]. Although all six PCGF proteins have highly similar RING domains, the variants of PRC1 complexes that they form vary considerably in their E3 ubiquitin ligase activity due to differences in cofactors. It is reported that these cofactors can specifically enhance the catalytic activity of PRC1 or promote its recruitment to nucleosomes [5]. Notably, ncPRC1 variants exhibit significantly stronger activity on nucleosome substrates than cPRC1 complexes, in which E3 ligase activity is significantly stimulated by the incorporation of the ncPRC1-specific accessory subunit RYBP [12,36].

### 2.2. PRC1 in Transcription Repression

Accumulating evidence suggests that PRC1 plays an important role in the repression of gene transcription through chromatin modifications [37,38]. Two mechanisms, H2AK119ub1, and chromatin compaction are currently proposed to account for the mediation of this gene silencing (Figure 2A–C) [7,39]. RING1A/B-catalyzed H2AK119ub1 provides a general mechanism for PRC1-mediated transcriptional repression. Genome-wide studies have shown a strong correlation between H2AK119ub1 deposition and gene inhibition [40,41,42]. Moreover, the occupancy of H2AK119ub1 on chromatin is able to maintain the conformational equilibrium of RNA polymerase II (RNAP) and inhibits its facilitating effect on transcription elongation, which supports the gene transcription repression function of PRC1 [41,43]. Phosphorylation of amino acid residues within the carboxy-terminal domain (CTD) of RNAP is associated with transcription initiation, elongation, and termination, where active transcription sites are typically characterized by phosphorylation of its Ser2 residues, whereas inactive or stable genes bind Ser5-phosphorylated RNAP in promoter-proximal regions [44]. H2AK119ub1 has been reported to inhibit the transcriptional activation of RNAP by balancing the recruitment of Ser5-phosphorylated RNAP and Ser2-phosphorylated RNAP at gene loci [41]. On the other hand, it can block the binding of RNAP at the early stages of elongation by preventing the recruitment of Facilitates Chromatin Transcription (FACT) to the promoter region of transcription [43]. In addition to its ability to counteract RNAP binding and transcription initiation, PRC1 may also play a critical role at promoters, controlling the frequency of transcriptional bursts to drive Polycomb-mediated gene repression by counteracting low-level or inappropriate transcriptional signals emanating from regulatory elements such as enhancers [45]. Furthermore, PRC1 can induce chromatin compaction through disordered regions of highly positively charged amino acids in its Psc subunits while repressing the transcription of genes [46].

PRC1 often cooperates closely with PRC2 to target, establish, and maintain transcriptional repression of PcG-targeted genes (Figure 2A,B) [47,48]. This cooperation was initially thought to be initiated by PRC2-catalyzed H3K27me3, which is then recognized by the CBX proteins of PRC1, driving cPRC1 to pre-occupied loci of PRC2 to exert transcriptional repression (Figure 2A) [6,47,49]. However, because there are no CBX proteins in ncPRC1, this model can only be applied to cPRC1 recruitment. Indeed, in PRC2-deficient mouse embryonic stem cells (ESCs), RING1B can also occupy the majority of gene loci targeted by PcG, suggesting that PRC1 can be recruited to nucleosomes in an H3K27me3-independent manner (Figure 2B) [33,50]. In addition, recent studies have complemented the mechanism of cooperation of ncPRC1 with PRC2, and researchers have shown that ncPRC1-mediated H2AK119ub1 recruits PRC2 [35,51]. Similar to the mechanism by which PRC2 recruits cPRC1, H2AK119ub1 catalyzed by ncPRC1 can be recognized and bound by JARID2, driving PRC2 to catalyze H3K27me3 at PcG-targeted gene loci to exert transcriptional repression [52,53].

PRC1 can also restrict gene transcription in a chromatin compaction manner, independent of histone H2A ubiquitination (Figure 2C) [32,39,54]. Preliminary in vitro analyses with short nucleosome arrays suggest that core components of the *D. melanogaster* PRC1, especially the Psc subunits, generate a compact chromatin structure by a mechanism involving interactions with nucleosomes without the need for histone tails [32,39]. Subsequent studies on ESCs with loss-of-function mutations in mouse RING1B (I53A) identified chromatin decompaction, and the addition of RING1B restored chromatin compaction in vivo, providing further evidence of a chromatin compaction role for PRC1 [54]. Furthermore, characterization of the organization of PcG-targeted genes in ESCs and neural progenitor cells using 5C and super-resolution microscopy revealed that chromatin compaction at PRC1-repressed loci formed isolated self-interacting domains and that chromatin compaction was only associated with cPRC1 [55]. The Psc subunits of the PRC1 complex play a key role in chromatin compaction in Drosophila, consistent with the predominant role of CBX proteins in mammals. Similar to Psc, CBX2 also has a disordered region of highly positively charged amino acids that is critical for inducing chromatin compaction in vitro [46,56]. Importantly, in a mouse model carrying CBX2 with a mutant nucleosome compaction region, homologous heterogeneous transformations similar to those observed with PcG loss-of-function mutations were observed, suggesting that CBX2-driven nucleosome compaction is a key mechanism by which PRCs maintain transcriptional repression during mouse development [57].

### 2.3. PRC1 in Transcription Activation

Traditionally, PRC1 is well known for its transcriptional repressive role. However, recent studies have found that PRC1 also functions as a transcriptional activator [58,59,60,61,62,63]. Genome-wide research of murine megakaryoblastic cells identified the PRC1 core component RING1B as sharing a large number of target gene loci with the transcription factor complex Runx1/CBFβ, and interestingly, knockdown of RING1B led not only to upregulation of target genes, but also to downregulation of target genes, suggesting that PRC1 might play a role in both transcriptional repression and transcriptional activation [64]. Subsequent studies of various mammalian cell types have implicated multiple components of PRC1, including PCGF1, CBX8, and RING1B in transcriptional activation of target genes [62,65,66,67]. However, the detailed molecular mechanisms of PRC1-mediated transcriptional activation are not fully understood as yet. Currently, the identified molecular mechanisms of PRC1 involved in transcriptional activation mainly include two aspects: the synergistic effect of PRC1 with other epigenetic regulators and the inhibition of intrinsic ubiquitination activity of PRC1 (Figure 2D,E) [58,59,60].

Studies in mouse neural progenitor cells have demonstrated that the AUTS2 subunit in ncPRC1.5 can recruit P300, a transcriptional coactivator and histone acetyltransferase that promotes acetylation of histone H3 at Lys27 (H3K27ac), facilitating transcriptional activation of genes [60]. In addition, Zhao et al. used a proteomic approach and promoter occupancy analysis to identify several novel PCGF3/5-interacting proteins, including testis expressed 10 (Tex10), which can directly contribute to transcriptional activation through P300 (Figure 2D) [59]. Furthermore, depletion of PCGF3/5 in ESCs significantly reduced the occupancy of Tex10 and P300 on target gene loci, indicating that PCGF3/5 acted as transcriptional activators through the interaction of Tex10 and P300 [59]. The mechanism by which P300 cooperates with ncPRC1.3/5 to activate gene transcription has been further confirmed in more recent studies [68,69].

As mentioned above, the E3 ubiquitin ligase activity of RING1B can be enhanced by several other PRC1 components, thereby promoting gene repression. In contrast, factors that inhibit RING1B-mediated H2AK119ub1 have been identified, which provides new insights into the mechanism of PRC1-mediated transcriptional activation [58,60]. Genome-wide ChIP-sequencing analysis (ChIP-seq) of mouse quiescent lymphoid B cells revealed that Aurora B kinase and cPRC1.4 colocalize at active promoters and they are required for RNAP binding to active promoters [58]. In addition, Aurora B kinase not only inhibits H2AK119ub1 by phosphorylating and inactivating the E2 enzyme UBE2D3 but also promotes H2A deubiquitination by phosphorylating and enhancing the activity of the deubiquitinating enzyme USP16 (Figure 2E, top) [58]. Further research subsequently uncovered an alternative mechanism for the inhibition of H2A mono-ubiquitination (Figure 2E, bottom). Interestingly, ncPRC1.3/5 tends to localize to gene loci lacking H2AK119ub1 compared to other PRC1 complexes, and in vitro reporter assays have identified a role for ncPRC1.3/5 in transcriptional activation [60]. Indeed, the AUTS2 subunit of the ncPRC1.5 complex recruited CK2, which then promoted gene transcriptional activation by binding to and phosphorylating Ser168 of RING1B, leading to a decrease in the E3 enzymatic activity of RING1B [60]. The reduction in PRC1 activity not only inhibits PRC2-mediated gene silencing, it also leads to the upregulation of transcriptional regulators that in turn activate target genes. For example, the reduction of H2AK119ub1 occupancy on chromatin disrupts the conformational equilibrium of RNAP, with an increase in Ser2-phosphorylated RNAP and a decrease in Ser5-phosphorylated RNAP, promoting its transcriptional initiation and elongation effects [41]. Decreased PRC1 activity would also deregulate low-level or inappropriate transcriptional signals from enhancers, thus promoting a transcriptional burst, rather than facilitating gene activation [45].

## 3. Composition of PRC2 and the Role It Plays in Transcription

PRC2 has generally been considered the relatively smaller of the two PRCs that have fewer PcG auxiliary subunits, but it is now becoming increasingly clear that, like PRC1, it also does not only exist as a single entity. Affinity purification coupled with tandem mass spectrometry (AP-MS) studies using a single PRC2 subunit as bait revealed that PRC2 comprises at least two distinct functional subcomplexes, referred to as PRC2.1 and PRC2.2 (Figure 1B) [70]. Here, we elaborate the composition of these two complexes and the function of these constituent subunits in PRC2-mediated transcriptional regulation.

### 3.1. Composition of PRC2 and Its Methyltransferase Activity

PRC2 is composed of four core subunits, EZH2 or its homologs EZH1, EED, SUZ12, and RBAP46/48 (also known as RBBP4/7), and is divided into PRC2.1 and PRC2.2, depending on their auxiliary subunits [71,72,73]. PRC2.1 includes a Polycomb-like (PCL) protein (PCL1/2/3), as well as Elongin BC and Polycomb repressive complex 2-associated protein (EPOP) or PRC2-associated LCOR isoform 1/2 (PALI1/2), whereas PRC2.2 includes Adipocyte enhancer binding protein 2 (AEBP2) and Jumanji and AT-rich interaction domain containing 2 (JARID2) [70,74,75,76].

PRC2 promotes chromatin compaction and gene silencing mainly through methylation (mono-, di-, and trimethylation) of H3K27 catalyzed by the enzyme subunits EZH1/2 [73,77]. Methylation of H3K27 is progressive (H3K27me3 is the result of mono-methylation of H3K27me2), and H3K27me3 is a stable mark [78]. In contrast to H3K27me3, the importance of H3K27me2 in maintaining gene repression appears limited [79]. However, H3K27me2 is an important intermediary PRC2 product that not only constitutes a substrate for subsequent H3K27me3 formation but may also prevent H3K27 from being acetylated. Acetylated H3K27 is thought to be antagonistic to PcG-mediated gene silencing and is enriched in the absence of PRC2 [80]. Unlike H3K27me2/3, H3K27me1 is still detectable in cells carrying non-functional PRC2 and its enrichment correlates with actively transcribed genes [81]. Exactly how H3K27me1 is generated is still an issue of debate. H3K27me1 may be catalyzed by PRC2, while its presence in actively transcribed genes also results from the demethylation of H3K27me2/3 by the demethylases lysine demethylase 6B (KDM6B) or (UTX histone demethylase) UTX [82].

Structural and biochemical analyses targeting PRC2 have focused on EZH2-containing complexes, which in most cases exhibit a more pronounced catalytic function for H3K27 methylation [73]. In PRC2, the stability of the core subunits EED, SUZ12, EZH2, and RBAP46/48 is strongly interdependent, and the steady state of the core complex is also necessary for PRC2 to function as a methyltransferase [83,84,85]. After EZH2 trimethylates H3K27, EED binds to H3K27me3 through its WD40 domain, causing the SET (Su(var)3-9, Enhancer-of-Zeste and Trithorax) domain of EZH2 to conform into the active conformation, allosterically activating its catalytic activity [86]. In addition, PRC2 can also spread to neighboring nucleosomes to exert methyltransferase activity by binding H3K27me3 via EED subunits [87,88,89]. SUZ12 associates with EZH2 and EED through its VEFS (VRN2-EMF2-FIS2-SUZ12) domain to promote PRC2 stabilization, and its interaction with PCL proteins play an important role in recruiting PRC2 to chromatin [90,91,92]. RBAP46/48 proteins can associate via its helix 1 with free histone H4, H3-H4 dimers, and tetramers, consistent with being required for PRC2 binding to unmodified nucleosomes, and they are also essential for PRC2 to fully exert its methyltransferase activity [83,93,94]. Importantly, although PRC2.1-specific and PRC2.2-specific subunits are not essential for PRC2 methyltransferase activity, recent studies have found that they also affect PRC2 activity (Figure 3) [95,96,97].

### 3.2. PRC2 in Transcription Regulation

PRC2-catalyzed H3K27me3 is generally considered a hallmark of gene silencing. In the field of developmental biology, there are two main types of H3K27me3-marked genes: (1) genes with both H3K27me3 and H3K4me3 are denoted “bivalent genes”, which are OFF but can be rapidly activated under subsequent cues [98]; (2) genes that have H3K27me3 and H2AK119ub1, which are OFF and appear to be more difficult to activate at subsequent stages.

Unlike PRC1, the catalytic activity of PRC2 is dependent on the presence and integrity of only four core subunits, and deletion of any one core subunit results in loss of catalytic activity with the remaining auxiliary subunits being nonessential for PRC2 activity but associated with their recruitment to chromatin and regulation of catalytic activity [6,71,72,96,97,99,100]. PRC2 is usually divided into two variants, PRC2.1 and PRC2.2, of which PRC2.1 can be further subdivided into EPOP-containing and PALI-containing PRC2 variants depending on the dissimilarity of its accessory subunits (Figure 1B) [5]. Both EPOP and PALI1/2 proteins individually interact with core PRC2 subunits, but their regulatory effects on PRC2 transcriptional repression function are mutually exclusive, with the presence of PALI1/2 enhancing its methylation enzymatic activity [70,74,97]. EPOP bridges PRC2.1 to Elongin B/C (EloB/C), which interacts with Elongin A (EloA) and promotes RNAP elongation, further inhibiting PRC2-mediated transcriptional repression (Figure 3A, bottom) [75,101]. In addition, both PRC2.1 variants have three PCL proteins: PCL1 (also referred to as PHF1), PCL2 (also referred to as MTF2), and PCL3 (also referred to as PHF19), which promote PRC2.1 recruitment to PcG-targeted gene loci and enhanced catalytic activity (Figure 3A, top) [79,102,103,104,105,106]. Of these, PCL2 promotes de novo recruitment of PRC2.1 by binding to unmethylated CpG islands, whereas PCL1/3 mediates PRC2.1 recruitment to chromatin by recognizing and binding H3K36me3 [106,107,108,109,110]. In particular, PCL1 can increase the residence time of PRC2.1 on chromatin, increasing H3K27me3 deposition at PcG-targeted gene loci [111]. Furthermore, PCL3 can recruit the demethylase NO66 for H3K27me3, helping to promote H3K36me3 removal and H3K27me3 deposition, thereby enhancing PRC2.1-mediated transcriptional repression [102].

Within PRC2.2, a core component of PRC2 associates with AEBP2 and JARID2, which synergistically and drastically increase the catalytic activity of EZH2 [112,113,114]. Recent studies have shown that the catalytic stimulation of EZH2 by JARID2 is partially mediated by the trimethylation of JARID2 at Lys116 (JARID2-K116me3): PRC2 first catalyzes JARID2-K116me3, then the binding of JARID2-K116me3 to the WD40 domain of EED, induces a conformational change of EZH2, leading to the increased catalytic activity of PRC2.2 (Figure 3B, top) [95,115]. However, the stimulation of EZH2 catalytic activity by AEBP2 is not fully understood, possibly because it increases the stability of PRC2.2, which increases EZH2 catalytic activity [116]. Moreover, AEBP2 stimulated PRC2.2 binding to nucleosomes by binding H2AK119ub1, thereby enhancing EZH2-mediated H3K27me3 deposition [52,96]. Interestingly, EZH inhibitory protein (EZHIP), a protein expressed predominantly in the gonads, was recently shown to interact with the allosterically activated PRC2.2 and inhibited its methyltransferase activity (Figure 3B, bottom) [117,118,119,120].

Collectively, the transcriptional repression exerted by PRC2 through catalyzing H3K27me3 requires the stable existing forms of the four core components, while other specific accessory subunits are also very important for its recruitment on chromatin and regulation of methyltransferase activity.

## 4. Polycomb Repressive Complexes in Cancer

Although gene repression mediated by the cooperation of PRC1 with PRC2 can essentially fully explain the role of PRCs in cancer, the sheer amount of data on these genes has been difficult to fully research, and thus the recent focus has shifted to investigate the roles played by subunits of PRCs in the context of cancer. Multiple components of PRCs play important roles in a variety of cancers, and they not only are implicated in cancer initiation and development but also function as tumor suppressors (Table 1 and Table 2).

### 4.1. Oncogenic Role of PRC1

PCGF4 (also known as BMI-1), the best-studied PRC1 gene in cancer, was one of the first PcG genes found in mammals and has been defined as a proto-oncogene that cooperates with the c-Myc oncoprotein to promote tumorigenesis [232,233,234,235]. The c-Myc protein can promote the transcription of BMI-1, thereby enhancing the transcriptional repression of genes such as p16 and p19^ARF^, which are tumor suppressors encoded by the *Ink4a/ARF* locus (Figure 4A) [131,236]. In addition, BMI-1 is overexpressed in gastric, colorectal, ovarian, breast, and other cancers, promoting cell proliferation and immortality by repressing transcription of the *Ink4a/ARF* locus [131,132,237,238,239]. However, silencing of this locus is unlikely to be the only mechanism by which it exerts its oncogenic effects, and BMI-1 has been found to promote stem cell expansion and tumorigenesis in an *Ink4a/ARF* independent manner in some cancers [28,135]. Post-translational modifications (PTMs) of BMI-1 have been reported to enhance its oncogenic effects. O-GlcNAcylation of BMI-1 at Ser255 mediated by O-GlcNAc transferase (OGT) increased the stability of the protein and its oncogenic activity (Figure 4B) [240]. Meanwhile, other homologous proteins of PCGF4 such as PCGF1, PCGF3, and PCGF6 have also been described to play oncogenic roles in various cancers (Figure 4C). Among them, PCGF1 was found to enhance stemness and promote cell proliferation in colorectal cancer (CRC) cells by activating the expression of genes including *CD133*, *CD44*, and *ALDH1A1* [164]. PCGF3 was shown to promote proliferation and migration of non-small cell lung cancer (NSCLC) through the PI3K/Akt signaling pathway, while mutations in PCGF6 enhanced breast cancer cell migration and metastasis by upregulating the expression of epithelial-mesenchymal transition (EMT)-related genes [166,167].

Moreover, RING1B (also known as RNF2), the core catalytic subunit of PRC1, was found to be overexpressed and promoted oncogene expression [25,121]. It has been reported to have a dual role of gene repression and activation, which drove cell proliferation by activating the expression of *CCND2* and promoted cell invasion by inhibiting the expression of *LTBP2* (Figure 4D) [24]. Additionally, RING1B has also been shown to promote EMT and metastatic progression of cancer cells by activating the expression of *ZEB2* and inhibiting the expression of *E-cadherin*, further demonstrating its important role in cancer development (Figure 4D) [24,122]. Recently, RING1B has been described to play multiple functions in basal-like and luminal breast cancer, where it elevated enhancer activity and promoted gene transcription [123]. Interestingly, RING1A and RING1B can also play oncogenic roles by denaturing other non-histone substrates, and both of them promote p53 protein degradation in colorectal, hepatocellular, and germ cell tumors [126,127]. In addition, RING1B can negatively regulate autophagy by binding Lys45 (K45) of DCAF3 in mouse embryonic fibroblasts (MEFs), which may also be implicated in exerting its oncogenic effects, but the specific mechanism remains to be further explored [241]. Not just the core subunits, but also numerous accessory subunits of PRC1 have also been shown to primarily play oncogenic roles (for a detailed description, see Table 1).

### 4.2. Tumor-Suppressive Role of PRC1

Generally, PRC1 is known for its oncogenic role. However, several studies have found that its components also exert tumor-suppressive effects. Most notably, unlike other PCGF homologs, PCGF2 (also known as MEL-18) has tumor-suppressive activity [31,128,129,242,243]. The expression levels of MEL-18 and BMI-1 are negatively correlated in various cancers [30,130]. Overexpression of MEL-18 leads to downregulation of c-Myc protein, a transcriptional activator of BMI-1, resulting in decreased levels of BMI-1 protein, promoting p16 and p19^ARF^ upregulation, ultimately inhibiting cell proliferation and driving cellular senescence (Figure 4A) [236]. Meanwhile, MEL-18 also negatively regulated the ubiquitination activity of RING1B by inhibiting BMI-1 transcription [244]. In addition, previous studies have also shown that MEL-18 loss causes aggressive phenotypes in breast cancer, such as facilitating stem cell activity, angiogenesis, cell cycle progression, and EMT (Figure 4E) [31,128,129,243]. More recently, MEL-18 loss was described as mediating estrogen receptor-α (ER-α) downregulation, resulting in a hormone-independent phenotype of breast cancer associated with poor prognosis, implicating a key role in the hormonal regulation of breast cancer [245]. PHC3, another component of cPRC1, is also known to act as a tumor suppressor, and its frequent loss of heterozygosity (LoH) in osteosarcoma promotes tumorigenesis [140,141].

Importantly, multiple accessory subunits also act as tumor suppressors, such as CBX proteins, RYBP, KDM2B, and BCOR (Table 1). Among them, the CBX4, CBX6, and CBX7 proteins have dual roles as oncogenic and tumor suppressors. CBX proteins mainly regulate cell migration, proliferation, and EMT, but interestingly in different tumors, they play quite opposite roles [29,146,147,148,151,153]. This may be because the auxiliary subunits of PRC1 are dynamically assembled into different PRC1 in a context-dependent manner, and thus play different or even opposite roles in the occurrence and development of cancer, but the specific mechanisms need to be further investigated clearly [246]. Unlike the CBX proteins, RYBP plays a predominantly tumor-suppressive role and is oncogenic in only a few cancers [168,169].

In summary, whether PRC1 plays an oncogenic or tumor-suppressive role is mainly related to its function in cell proliferation, cell cycle progression, metastasis, and invasion (References are shown in Table 1). Given the prominent oncogenic role that PRC1 plays, inhibitors targeting PRC1 for antitumor therapy have also attracted attention, and a subset of inhibitors have been reported [247,248,249,250]. However, the precise roles of the various subunits of PRC1 in cancer remain to be defined, and future work should further delineate the molecular implications of these components in depth and identify appropriate therapeutic approaches to rescue their dysregulation in different cancers.

### 4.3. Oncogenic Role of PRC2

Currently, it has been reported that the pro-tumor activity of PRC2 is mainly associated with its EZH2, EED, and SUZ12 subunits. The first clinically relevant finding in the PcG protein field was that EZH2 promoted prostate cancer progression and poor prognosis [15]. It was subsequently identified as being downstream of the pRB-E2F pathway, which was essential for tumor cell proliferation [17]. Dysregulation of EZH2, as well as its roles, has been discussed in several solid malignancies including prostate, hepatocellular, colorectal, and breast cancer, as well as in some hematologic malignancies [251]. EZH2 is involved in cancer initiation and progression mainly due to its transcriptional repression activity in the PRC2 complex, and gain-of-function (GOF) mutants of EZH2 (Y647F/N, A677G, and A687V) are frequently generated in several lymphomas, further promoting tumorigenesis (Figure 5A) [19,20,189]. Similar to BMI1, PTMs of EZH2 have also been reported to enhance its oncogenic effects. Acetylation at Lys348 (K348) and O-GlcNAcylation at Ser73 (S73) increased the protein stability and catalytic activity of EZH2, further enhancing its ability to promote cancer cell migration and invasion (Figure 5B) [252,253]. In particular, EZH2 may exert oncogenic effects not only by mediating H3K27me3, but also by methylating non-histone proteins or binding to other proteins in a PRC2-independent manner. Recent studies have found that EZH2 can methylate the transcription factor RORα at Lys38 (K38), promoting the degradation of RORα and reducing RORα-mediated activation of gene transcription. Interestingly, the levels of EZH2 and RORα were inversely correlated in breast cancer, implying that EZH2 might play an oncogenic role by inhibiting RORα-mediated tumor suppression [254]. In contrast, phosphorylated EZH2 methylates the transcription factor STAT3 at Lys180 (K180), which enhances STAT3-mediated transcriptional activation but also promotes cancer stem cell self-renewal and exerts oncogenic effects (Figure 5C) [255]. Moreover, EZH2 is able to directly interact with β-catenin and ERα in a PRC2-independent manner to activate *cyclin D1* and *c-Myc* expression upon estrogen stimulation to induce the proliferation of breast cancer cells (Figure 5D) [256]. Comparably, the β-catenin activation complex can also recruit EZH2 via PCNA associating factor (PAF) to activate the Wnt signaling pathway that drives tumorigenesis (Figure 5D) [257]. However, the specific mechanism by which EZH2 activates gene expression in a PRC2-independent manner remains unclear. Taken together, EZH2, whether acting in a PRC2-dependent or PRC2-independent manner for gene regulation, has been described to be closely associated with tumorigenesis and progression.

Additionally, EED and SUZ12 are upregulated in multiple cancers, including lymphoma, breast cancer, head, and neck squamous cell carcinoma (NHSCC), and colorectal cancer [200,201,206]. The knockdown of SUZ12 significantly inhibited cell proliferation, invasion, and migration in HNSCC cells and inhibited xenograft tumor growth [206]. However, the upregulation of EED and SUZ12 is often accompanied by the upregulation of EZH2, exerting oncogenic effects mainly through PRC2-mediated gene silencing [200,201]. In addition, the oncogenic roles played by other subunits of PRC2, such as PCL1/2/3 and EPOP have also been linked to their regulation of PRC2 activity or promotion of PRC2 recruitment to chromatin (Table 2) [75,215,219,222]. In contrast, RBAP46 bound the transcription factor Sp1 in a PRC2-independent manner, resulting in the downregulation of reversion-inducing cysteine-rich protein with Kazal motifs (RECK), a protein that suppresses tumor metastasis and angiogenesis [213].

### 4.4. Tumor-Suppressive Role of PRC2

Although PRC2 has oncogenic properties in most tumors, it has been shown that EZH2 and other PRC2 subunits also have tumor-suppressive functions in some types of tumors [23,191,209,230]. In contrast to GOF mutants, loss-of-function (LOF) mutants of EZH2 have also been identified in T-cell acute lymphoblastic leukemia (T-ALL). LOF mutants of EZH2 (G266E, T393M, and C606Y) that result in loss of PRC2 function and drive Notch signaling activation, and increase the in vivo tumorigenic potential of T-ALL cells, suggesting that PRC2 may have a tumor-suppressor function, although the specific mechanisms remain to be further explored (Figure 5A) [192]. Recently, PTMs of EZH2 have also been shown to exert tumor-suppressive effects. Methylation at Lys735 (K735) and Phosphorylation at Thr261 (T261) increased the protein stability and catalytic activity of EZH2, further attenuating its transcriptional repression on tumor suppressors (Figure 5B) [258,259]. Moreover, in malignant peripheral nerve sheath tumors (MPNSTs), there is the frequent deletion of PRC2 subunit genes, which also leads to loss of H3K27me2 and H3K27me3 and is associated with poor prognosis [23,260] On the other hand, by exogenously expressing the deleted PRC2 subunit in MPNST cells with frequent deletion of the PRC2 gene, H3K27me3 levels were increased and cell proliferation was inhibited [23]. Furthermore, treatment with EZH2 inhibitors had no effect on the proliferation of MPNST cells [198]. Similarly, SUZ12 loss in T-ALL, which results in decreased gene silencing function of the PRC2 complex, promotes oncogene upregulation [261]. These results further support that PRC2 has tumor suppressor properties in specific cancers.

Interestingly, deletion of EED in mice resulted in hyperproliferation of myeloid progenitors and lymphoid cells, which accelerated lymphoid tumor formation after exposure of mice to genotoxic drugs, although this did not induce tumorigenesis [262,263]. Likewise, EZH2 or SUZ12 deletion accelerated Myc-driven lymphomagenesis by limiting self-renewal of B cell progenitors [264]. In addition, EZH2 loss significantly promoted the development of myelodysplastic syndrome induced by transcription factor Runx1 mutation, and loss of SUZ12 synergizes with neurofibromin 1 (NF1) mutations to amplify Ras signaling to drive cancer [210,265,266].

In conclusion, in addition to the partial non-canonical function of EZH2, PRC2 mainly exerts oncogenic or tumor-suppressive effects through its gene silencing function, depending on the cancer type. Both enhancement and attenuation of PRC2 catalytic activity can promote tumor development, suggesting that PcG genes are context-dependent tumor suppressors or oncogenes. Further observations of the context-dependent roles of PRC2 have revealed that the effects of loss-of-function and gain-of-function alterations do not simply segregate based on tissue or tumor type [267]. Furthermore, several studies have shown that the role of PRC2 in cancer depends on tumorigenic alterations in other genes [268,269]. For example, in NSCLC with loss-of-function mutations in BRG1 (W764R) or gain-of-function mutations in EGFR (T790M and L858R), inhibition of EZH2 catalytic activity promotes apoptosis and sensitivity to topoisomerase II (TopoII) inhibitors. Conversely, in BRG1 wild-type tumors, inhibition of EZH2 upregulates BRG1 and eventually confers stronger resistance to TopoII inhibitors [268]. Furthermore, in a *Kras*-driven mouse model of NSCLC, *EED* loss accelerated or delayed tumor formation depending on p53. In a WT-p53 background, *EED* loss promotes inflammation, whereas p53 inactivation leads to invasive mucinous adenocarcinoma [269]. Thus, the context-dependent role of PRC2 suggests that its function in specific cancer types is enormously complex and future work will be beneficial to exhaustively characterize its molecular implications in different cancers that will also help in identifying appropriate approaches to reverse their deregulation in different cells and provide suitable therapies for PRC2-dependent cancers. It is also notable that certain tumors are addicted to specific PRC2 subunits, independent of other components, the reason for which also remains ambiguous. Clearly, a better understanding of the cell-type-specific functions of each PRC2 subunit will require future research.

### 4.5. Development of Inhibitors Targeting Polycomb Repressive Complexes

PcG components have been reported to be associated with the growth and survival of different tumors, considered as targets for cancer therapy, and extensively explored [270,271]. Several PRC1-related inhibitors including PRT4165, PTC-209, IFM-11958, and RB-3 have been reported to date, but no inhibitor has entered clinical trials (Figure 6) [247,248,249,250]. PRT4165 was shown to inhibit the E3 enzymatic activity of PRC1, whereas PTC-209 and IFM-11958 were described as inhibitors of the BMI-1 expression. Furthermore, PTC-209 treatment significantly inhibits proliferation and promotes apoptosis in multiple myeloma (MM) cells, suggesting that BMI-1 may serve as an attractive anti-tumor drug target [248]. However, certain questions remain to be addressed regarding these inhibitors. Although PRT4165 was shown to be able to inhibit PRC1-mediated H2A ubiquitination, the mechanism by which it directly or indirectly inhibits the catalytic activity of PRC1 is not well elucidated. PTC-209 and IFM-11958 have been described to exert antitumor effects by inhibiting BMI-1 expression, but whether this effect is through inhibition of the function of PRC1 or other PRC1-independent functions of BMI-1 is unclear. Therefore, there is a critical need to develop inhibitors that specifically target PRC1. RB-3 was subsequently developed to inhibit the binding of RING1B and BMI-1, while specifically inhibiting the catalytic activity of PRC1 [247]. Importantly, RB-3 treatment drastically reduced the global level of ubiquitination of H2A and induced differentiation of leukemia cell lines, implying that inhibitors of PRC1 might be used to carry out the treatment of leukemia [247]. However, although numerous accessory subunits of PRC1 are oncogenic, PRC1 exerts oncogenic functions not only by repressing tumor suppressors but also by activating oncogenes, so it remains to be explored whether PRC1 components are suitable therapeutic targets [40,67,123,131].

Unlike PRC1, PRC2 exerts oncogenic effects mainly dependent on its transcriptional repression properties [5]. Therefore, the development of an inhibitor that targets PRC2, either by inhibiting its methyltransferase activity or by interfering with the stability of the complexes, could be a promising strategy for the treatment of PRC2-dependent tumors [182,271]. One of the first therapeutic agents reported to target PRC2 was 3-deazaneplanocin A (DZNep), which reduces H3K27me3 levels and causes apoptosis in cancer cells [272,273]. However, subsequent studies showed that instead of specifically inhibiting PRC2, DZNep reduced H3K27me3 levels by altering other histone methylation (H4K20me3) levels [274]. Medicinal chemists then have developed EZH2-specific enzymatic inhibitors including GSK126 and EPZ005687, both of them act by binding competitively to the SET domain of EZH2 and they have shown better selectivity and greater potency compared to DZNep [188,275]. Furthermore, both compounds not only target wild-type EZH2, but are also extremely sensitive to lymphoma cells harboring EZH2-activating mutations [188,275]. To date, multiple inhibitors targeting EZH2 have been reported, of which Tazemetostat was approved by the Food and Drug Administration (FDA) in January 2020 for the treatment of advanced epithelioid sarcoma and follicular lymphoma [276,277]. There are also many potent and more bioavailable EZH2 catalytic inhibitors currently undergoing phase 1/2/3 clinical trials, alone or in combination with other drugs, for the treatment of several solid tumors, mainly lymphoma, prostate cancer, and small cell lung cancer (Table 3). In addition, Tazemetostat is being used in several phase 2 clinical trials for the treatment of diffuse large B-cell lymphoma, hematologic neoplasms, mantle cell lymphoma, and peripheral nerve sheath tumors (NCT05205252, NCT04917042). Constellation pharmaceuticals has also tested multiple EZH2 inhibitors in clinical trials, including CPI-1205 and CPI-1205; CPI-1205 (NCT03480646), as their first-generation EZH2 inhibitor, is currently in phase 2 testing for metastatic castration-resistant prostate cancer, while a second-generation EZH2 inhibitor (CPI-0209) is also being tested in phase 2 for solid tumors (NCT04104776). An EZH2 inhibitor (PF-06821497) developed by Pfizer has entered phase 1 testing in patients with small cell lung cancer, castration-resistant prostate cancer, and follicular lymphoma (NCT03460977). As discussed earlier, the integrity and stability of PRC2 is essential for its methyltransferase activity, which provides additional possibilities for the development of inhibitors targeting PRC2. In addition to EZH2 catalytic inhibitors, an alternative strategy is to inhibit the allosteric activation of EZH2 enzymatic activity by preventing the interaction of EED-H3K27me3 using small molecule inhibitors (Figure 6) [203,278,279]. A potential advantage of this approach is that the specificity of the EED-H3K27me3 interaction may increase the selectivity for inhibition of PRC2, whereas other SET-domain-containing methyltransferases may be targeted by poorly selective EZH2 enzymatic inhibitors. EED226 was one of the first EED inhibitors reported and was found to specifically inhibit H3K27 methylation, either in lymphoma cells harboring WT-EZH2 or EZH2 mutants [278]. Furthermore, gene expression microarray and Chip-PCR experiments revealed that PRC2-regulated gene expression was significantly increased by treatment with EED226 and was similar to that of EZH2 catalytic inhibitor (EI1) treatment, demonstrating that EED226 also has a significant inhibitory effect on the gene-silencing function of PRC2 [278]. Interestingly, although EED226 failed to enter clinical trials, another EED inhibitor, MAK683, developed on the basis of this scaffold, is currently in phase 2 clinical trials for the treatment of lymphoma (Table 3) [280]. Moreover, EEDi-5285 is the most potent EED inhibitor that has been reported, and it achieves complete and long-lasting tumor regression in mice, further suggesting great potential for the treatment of PRC2-dependent cancers by inhibiting the EED-H3K27me3 interaction [279]. Other approaches to inhibiting PRC2 activity in cancer include a staple peptide inhibitor that disrupts the protein-protein interaction (PPI) of EZH2-EED, with significant antiproliferative functions even in cells resistant to EZH2 inhibitors [281]. Subsequent studies reported that the FDA-approved drug astemizole and the natural compound wedelolactone could also inhibit PRC2 activity by blocking PPI of EZH2-EED, but these inhibitors were less potent than known EZH2 inhibitors or EED inhibitors [282,283].

## 5. Concluding Remarks and Perspective

PRCs have attracted increasing attention because of their important role in a wide range of physiological and pathological processes, especially in cancer treatment (Table 1 and Table 2). The gradually discovered new members of PcG in the complexity of PRCs open new perspectives for research in this field. In addition, the identification and characterization of multiple PRC variants have also provided a new understanding of the specific mechanisms by which PRC1 and PRC2 exert catalytic activity. In this review, we have discussed the currently identified PRC1 variants and PRC2 variants and classified them separately using a similar classification system (Figure 1). This classification is based on the relatively new but robust and cross-validated biochemical experimental results reported so far. Although components of different PRC variants have been reported, important aspects such as their abundance, stoichiometry, assembly of different variants, and an exact number of PRC variants formed in the cell remain to be addressed. Importantly, further exploration of the role of newly identified accessory subunits may have significant implications for complementing the function of PRC variants.

PcG proteins are involved in highly dynamic biological processes by regulating the transcription of genes. Among them, PRC1 is reported to have dual roles of gene silencing and activation, which may be related to its enormous complexity and a large number of subunits. It will be valuable to expend effort to uncover the molecular principles underlying PRCs gene regulation and decipher their mechanisms in Polycomb-related disorders. In particular, PcG proteins are hallmark components of cancer research, and many agents targeting PRCs have been developed and have advanced into clinical trials (Table 3). Considering that PRCs may play opposite functions in different cancers [284], understanding the specific role of each complex variant in a specific cancer type is even more critical and will help to develop effective therapeutic strategies.

Despite our established understanding of PRCs, it is clear that we are only beginning to unravel the possibilities behind these rather complex epigenetic complexes, with a plethora of PcG proteins whose functions in cancer have yet to be discovered and many questions remain to be addressed. Current studies have found both canonical and non-canonical activities and functions of PRCs in tumor initiation and progression, but what are their exact contributions? PMTs of multiple subunits of PRCs were shown to render them functionally altered, but what are the specific mechanisms that trigger the occurrence of these PMTs in cancer? PRCs have been reported to harbor a large number of variants, but is there functional crosstalk between these different variants in specific cancers? If present, what are the specific functional crosstalk mechanisms? The physiological, pathological, and therapeutic possibilities behind such epigenetic complexes remain to be revealed with significant effort. With the remarkable development of our research techniques and methods, the coming years will inevitably be fast-paced and exciting years to explore and define PRCs.

## Figures and Tables

**Figure 1 ijms-23-09574-f001:**
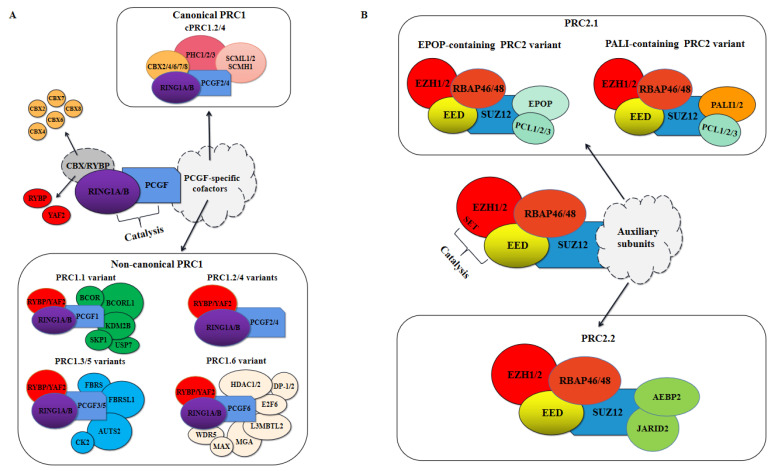
The classification of mammalian Polycomb repressive complexes. (**A**) The RING1A/B and Polycomb group RING finger proteins (PCGF1–6) are the catalytic core of PRC1 and they interact with a range of accessory subunits to create different PRC1 variants, mainly divided into canonical PRC1 (cRPC1) and non-canonical PRC1 (ncPRC1). cPRC1 complexes (**top**) assemble around PCGF2/4 and include Chromobox proteins (CBX2/4/6/7/8), Polyhomeotic protein (PHC1/2/3), and SCM proteins (SCML1/2 or SCMH1). By contrast, ncPRC1 complexes (**bottom**) can assemble around all six PCGF proteins and contain YY1-associated factor 2 (YAF2) and RING and YY1 binding protein (RYBP). Depending on its PCGF protein, it can be divided into PRC1.1–1.6 variants (also known as ncPRC1.1–1.6). (**B**) PRC2 is composed of four core subunits, EZH1/2, EED, SUZ12, and RBAP46/48 (also referred to as RBBP4/7), and is divided into PRC2.1 (**top**) and PRC2.2 (**bottom**) depending on their auxiliary subunits, of which PRC2.1 can be further subdivided into EPOP-containing and PALI-containing PRC2 variants (**top**).

**Figure 2 ijms-23-09574-f002:**
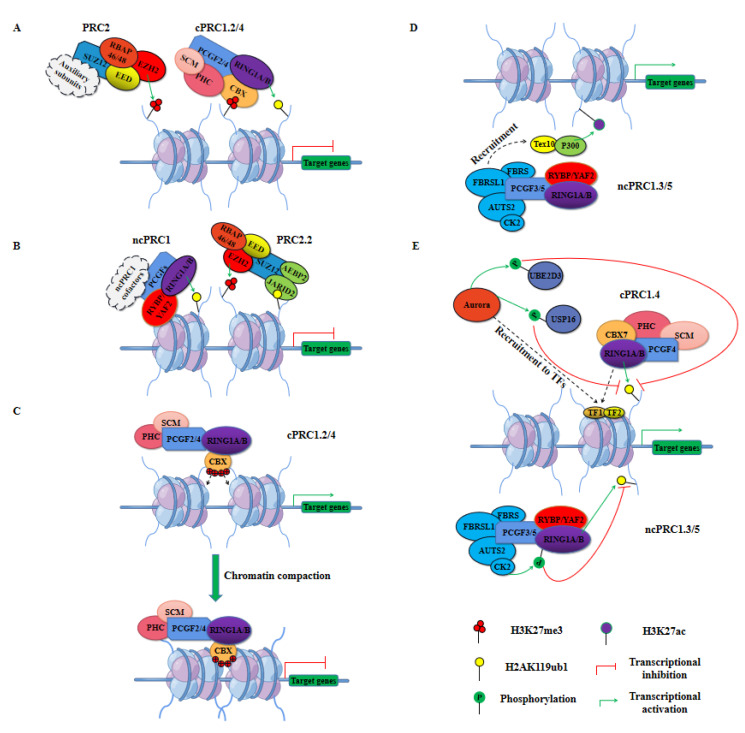
PRC1-mediated transcriptional repression and activation. (**A**) Crosstalk between cPRC1.2/4 and PRC2. PRC2 catalyzes trimethylation of histone H3 at Lys27 (H3K27me3), followed by CBX proteins recognition of H3K27me3, drives cPRC1.2/4 to PRC2 pre-occupied gene loci, and mediates mono-ubiquitination of histone H2A at Lys119 (H2AK119ub1), promoting transcriptional repression. (**B**) Crosstalk between ncPRC1 and PRC2.2. ncPRC1 first catalyzes H2AK119ub1, then PRC2 recognizes and occupies the same gene locus via JARID2, catalyzes H3K27me3, and promotes transcriptional repression. (**C**) The cPRC1.2/4 complexes mediate chromatin compaction to repress gene transcription, which is mediated by the interaction of the positively charged region of the CBX proteins with nucleosomes. (**D**) AUTS2 of ncPRC1.3/5 recruits Tex10 and P300 to acetylate histone H3 at Lys27 (H3K27ac), while promoting transcriptional activation. (**E**) Subunits or associated proteins of PRC1 directly or indirectly neutralize the E3 enzymatic activity of RING1A/B of PRC1, promoting transcriptional activation. CK2 inhibited its ubiquitination activity directly by phosphorylating RING1A/B, whereas Aurora kinases inhibited H2A ubiquitination indirectly by phosphorylating UBE2D3 and USP16.

**Figure 3 ijms-23-09574-f003:**
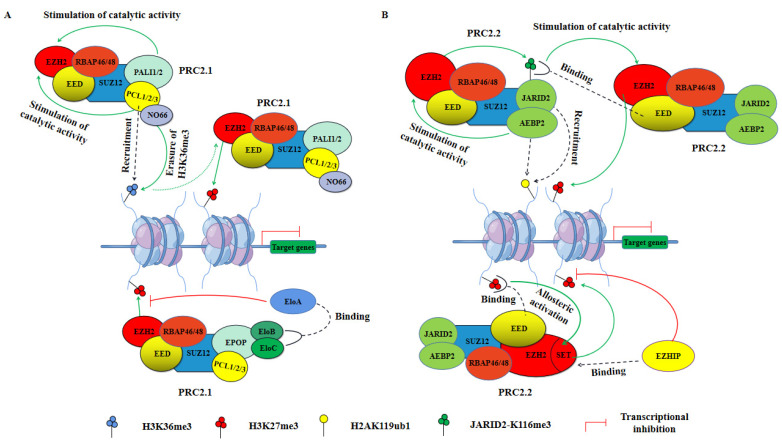
PRC2-mediated transcriptional regulation. (**A**) PRC2.1 in transcriptional regulation. All PCL and PALI1/2 proteins stimulate the methyltransferase activity of PRC2.1. PCL1/3 can bind to H3K36me3 and recruit PRC2.1 to gene loci containing high H3K36me3 levels. PCL2 recruits PRC2.1 to unmethylated CpG islands. Moreover, PCL3 can recruit the demethylase NO66 to erase the trimethylation modification of H3K36, allowing the deposition of H3K27me3 and further recruitment PRC2, promoting PcG-mediated transcriptional repression of target genes (**top**). EPOP associates with Elongin B/C (EloB/C), which binds Elongin A (EloA) and fine-tunes the transcriptional repression function of PRC2 on target genes (**bottom**). (**B**) PRC2.2 in transcriptional regulation. Both JARID2 and AEBP2 enhance the methyltransferase activity of EZH2 and can recruit PRC2.2 to PcG-targeted gene loci by binding H2AK119ub1. JARID2 is methylated at Lys116 by PRC2.2 and then recognized by EED, which allosterically activates the catalytic activity of EZH2, whereas AEBP2 stimulates the catalytic activity of EZH2 by increasing the stability of PRC2.2 (**top**). EED allosterically activates the catalytic activity of EZH2 by recognizing H3K27me3 and enhances PRC2.2-mediated transcriptional repression, whereas EZH inhibitory protein (EZHIP) can bind to allosterically activated PRC2.2 and inhibit its methyltransferase activity (**bottom**).

**Figure 4 ijms-23-09574-f004:**
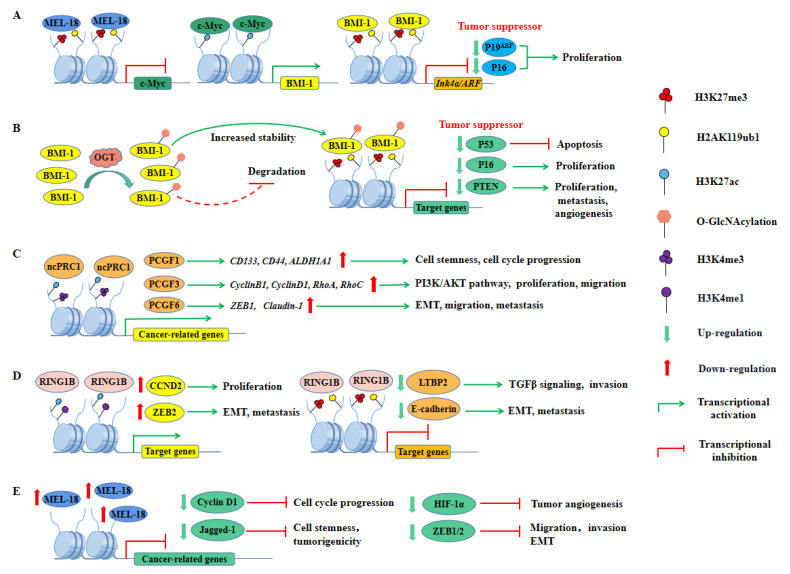
Multifaceted roles of PRC1 in cancer. (**A**) The c-Myc protein leads to increased PCGF4 (BMI-1) expression and promotes the transcription repression of target genes *p19^ARF^* and *p16* repressed at the *Ink4a/ARF* locus, promoting the proliferation of cancer cells. In contrast, PCGF2 (MEL-18) plays a tumor suppressor role by inhibiting the transcription of c-Myc. (**B**) PTMs of PRC1 subunits can promote tumorigenesis. Deposition of O-GlcNAcylation on BMI-1 improves its protein stability and inhibits its degradation. Increased levels of BMI-1 protein enhance transcriptional silencing of downstream target genes such as *p53*, *p16*, and *PTEN*, thereby promoting cancer cell proliferation, metastasis, and angiogenesis. (**C**) PRC1 may also be involved in tumorigenesis and development in a PRC2-independent manner. Interestingly, PRC1 is also found on specific targets that lack the H3K27me3 mark, and these gene loci deposit active marks such as H3K27ac and H3K4me3. The PRC1 subunits PCGF1, PCGF3, and PCGF6 were reported to activate related mechanisms such as PI3K/Akt pathway, EMT, proliferation, and metastasis, thus promoting tumorigenesis. (**D**) The catalytic subunit of PRC1, RING1B, promotes tumorigenesis through a dual role of gene repression and activation. It promotes cell proliferation and metastasis by activating the transcription of genes such as *CCND2* and *ZEB2*. Meanwhile, it can also promote cell proliferation, invasion, and metastasis by inhibiting the transcription of genes such as *LTBP2* and *E-cadherin*. (**E**) Overexpression of MEL-18 has been found to play a tumor suppressor role in a variety of cancers. MEL-18 negatively regulates cancer cell proliferation, angiogenesis, invasion, and metastasis by inhibiting the expression of genes such as *cyclin D1*, *Jagged-1*, *HIF-1α*, and *ZEB1/2*.

**Figure 5 ijms-23-09574-f005:**
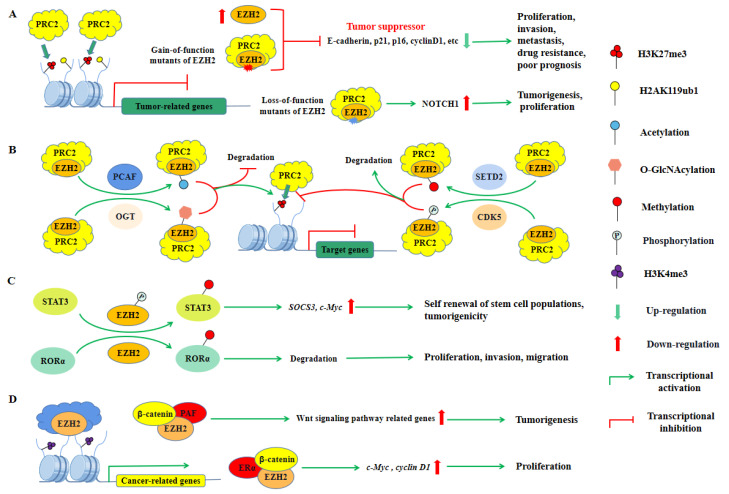
Multifaceted roles of PRC2 in cancer. (**A**) PRC2-mediated gene silencing was found to have both oncogenic and tumor-suppressive roles. Overexpression or gain-of-function mutations of EZH2 enhance the catalytic activity of PRC2 and its transcriptional inhibition of *E-cadherin*, *p21*, *p16*, *cyclinD1,* and other genes, thus promoting the growth, invasion, and metastasis of tumor cells, which is also related to drug resistance and poor prognosis. Surprisingly, loss-of-function mutants of EZH2 are also found in certain cancers and promote tumorigenesis, implicating the tumor suppressor role of PRC2 as well. (**B**) PTMs of EZH2 can promote or inhibit tumorigenesis. Deposition of O-GlcNAcylation and Acetylation on EZH2 inhibits its degradation and enhances the catalytic activity of PRC2. Conversely, deposition of methylation and phosphorylation on EZH2 promotes its degradation and inhibits the catalytic activity of PRC2. (**C**) Methylation of non-histone proteins mediated by EZH2 promotes tumorigenesis in a PRC2-independent manner. Methylation of RORα mediated by EZH2 promotes its degradation and promotes tumor cell proliferation, metastasis, and invasion. In addition, phosphorylated EZH2-mediated STAT3 methylation promotes the expression of *SOCS3*, *c-Myc,* and other genes, thereby promoting the self-renewal of stem cell populations (**D**) EZH2 binds to other proteins to promote the expression of tumor-related genes. EZH2 interacts with ERα and β-catenin and activates *c-Myc* and *cyclin D1* expression, which promotes cell proliferation. EZH2 interacts with ERα and β-catenin and stimulates the expression of Wnt pathway-related genes, thus promoting tumorigenesis.

**Figure 6 ijms-23-09574-f006:**
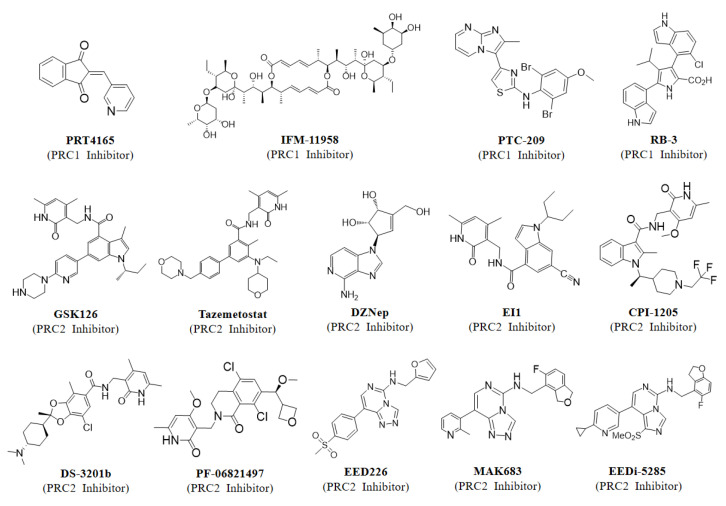
Inhibitor structures targeting PRC1 or PRC2.

**Table 1 ijms-23-09574-t001:** PRC1 components and their roles in cancer.

Complex ^a^	Subunit ^b^	Function	Descriptions	Cancer type	Refs.
cPRC1 or ncPRC1	RING1A	Oncogenic	Overexpression of RING1A promotes oncogene expression	Acute myeloid leukemia	[25]
	RING1B (RNF2)	Oncogenic	Overexpression of RING1B promotes oncogene expression	Acute myeloid leukemia,	[25,121,122]
			RING1B drives proliferation by upregulating transcription of cell cycle regulators	Melanoma	[24]
			RING1B not only promotes the expression of oncogenes but also regulates chromatin accessibility, and improves enhancer activity thus facilitating gene transcription	Breast cancer	[123,124,125]
			RING1B promotes p53 protein degradation	Colorectal cancer, Hepatocellular carcinoma	[126,127]
	PCGF2 (MEL-18)	Tumor suppressor	Overexpression of PCGF2 suppresses oncogene expression	Gastric cancer, Breast cancer	[31,128,129,130]
	PCGF4 (BMI-1)	Oncogenic	PCGF4 represses transcription of the *Ink4a/ARF* locus to promote cellular immortality	Non-small cell lung cancer Lymphoid, Breast cancer,	[131,132,133,134]
			PCGF4 promotes stem cell expansion and tumorigenicity in an *Ink4a/ARF* independent manner	Hepatocellular carcinoma, Glioma	[28,135]
			Ectopic expression of PCGF4 induces EMT and enhances tumor cell invasion and metastasis	Breast cancer, Nasopharyngeal cancer	[132,136,137]
			PCGF4 binds the androgen receptor (AR) and increases its stability, enhancing AR signaling in prostate cancer cells in a PRC1-independent manner	Prostate cancer	[138,139]
cPRC1	PHC3	Tumor suppressor	Missense mutations in PHC3 promote cell proliferation (G201C)	Osteosarcoma	[140,141]
cPRC1	CBX2	Oncogenic	Loss of CBX2 inhibits cell proliferation, invasion, and migration	Gastric cancer, Breast cancer	[142,143]
			Knockdown of CBX2 inhibits cell proliferation and promotes apoptosis	Ovarian cancer	[26]
	CBX4	Oncogenic	Promotes angiogenesis and metastasis of tumors	Hepatocellular carcinoma	[144,145]
			CBX4 significantly promotes tumors growth and metastasis	Clear cell renal cell carcinoma	[146]
		Tumor suppressor	Overexpression of CBX4 inhibits cell migration, invasion, and metastasis	Colorectal carcinoma	[147]
	CBX6	Oncogenic	Overexpression of CBX6 promotes EMT	Hepatocellular carcinoma	[148]
		Tumor suppressor	Exogenous overexpression of CBX6 inhibits cell proliferation, migration, and invasion, and induces cell cycle arrest	Breast cancer	[29]
	CBX7	Oncogenic	CBX7 represses transcription of the *Ink4a/ARF* locus and promotes tumorigenesis and invasion	Lymphoma, Prostate cell	[149,150]
			Overexpression of CBX7 promotes E-cadherin expression and is required for cell migration and invasion	Thyroid neoplasia, Cervical cancer	[151,152]
		Tumor suppressor	Reduced expression of CBX7 in cancer promotes cell progression and proliferation	Lung cancer, Bladder cancer	[153,154,155]
			CBX7 inhibits tumor proliferation by inactivating the tumor necrosis factor (TNF) signaling pathway	Clear cell renal cell carcinoma	[156]
	CBX8	Oncogenic	Premature senescence and growth arrest of cancer cells are suppressed by CBX8	Breast cancer, Leukemia	[157,158]
			Ectopic expression of CBX8 promoted tumor metastasis and growth, and overexpression of CBX8 in hepatocellular carcinoma cells activated Akt/β-catenin signaling	Hepatocellular carcinoma	[159,160]
cPRC1	CBX8	Oncogenic	CBX8 contributes to tumorigenesis or promotes stemness in specific tumors by acting non canonically	Mammary carcinoma, Colon cancer	[161,162]
ncPRC1	PCGF1	Oncogenic	Overexpression of PCGF1 promotes tumor cell cycle progression and cell proliferation	Cervical carcinoma	[163]
			PCGF1 activated stemness markers and promoted stem cell enrichment and self-renewal	Colorectal cancer, Malignant glioma	[164,165]
	PCGF3	Oncogenic	Overexpression of PCGF3 promoted cancer cell proliferation and migration.	Non-small cell lung cancer	[166]
	PCGF6	Oncogenic	PCGF6 promotes cell migration and metastasis by driving EMT	Breast cancer	[167]
	RYBP	Oncogenic	Silencing of RYBP inhibits melanoma cell proliferation, migration, and invasion	Melanoma	[168]
		Tumor suppressor	Overexpression of RYBP inhibits the degradation of tumor suppressor proteins and reduces cancer cell proliferation, migration, and metastasis	Breast cancer, Colon cancer, Lung cancer, Thyroid cancer, Hepatocellular carcinoma	[169,170,171,172]
	YAF2	Oncogenic	YAF2 is overexpressed in a variety of cancers and has been found to inhibit apoptosis	Non-small cell lung cancer, Breast cancer, Colon cancer	[173,174]
	KDM2B	Oncogenic	Overexpression of KDM2B Promotes the self-renewal of cancer stem cells	Breast cancer, Acute myeloid leukemia	[175,176]
		Tumor suppressor	Represses the expression of Notch pathway-related genes	T-ALL	[177]
	BCOR	Oncogenic	BCOR promotes PRC1-mediated dysregulation of expression through gene fusion	Endometrial stromal sarcoma	[178,179]
ncPRC1	BCOR	Tumor suppressor	Overexpression of BCOR inhibits cancer stem cell proliferation and self-renewal	T-ALL	[180]
	AUTS2	Oncogenic	The fusion PAX5-AUTS2 is a recurrent fusion gene in B-cell acute lymphoblastic leukemia	B-ALL	[181]

Complex ^a^, different variants of PRC1; Subunit ^b^, defined as a Polycomb group (PcG) protein present in the specific variant; Italicized format, gene name; Superscript numbers, references.

**Table 2 ijms-23-09574-t002:** PRC2 components and their roles in cancer.

Complex ^a^	Subunit ^b^	Function	Descriptions	Cancer type	Refs.
PRC2.1 or PRC2.2	EZH2	Oncogenic	EZH2 is overexpressed in tumors and promotes tumor cell proliferation and invasion	Prostate cancer, Breast cancer, Bladder cancer, Gastric cancer, Lymphoma, etc.	[15,16,182,183,184,185,186,187,188]
			Gain-of-function mutations in EZH2 increase the methylation levels of PcG-targeted genes and promote cell proliferation	Lymphoma	[18,20,189,190]
		Tumor suppressor	Loss-of-function (LOF) mutations or deletions of EZH2 promote oncogene expression	T-ALL, MPN	[191,192,193,194]
			EZH2 inhibits cell proliferation	Breast cancer, AML, B-ALL	[195,196,197]
	EZH1	Oncogenic	EZH1 and EZH2 are required for epithelial-mesenchymal transition (EMT) and cell proliferation	Breast cancer, MPNST	[198,199]
	EED	Oncogenic	Overexpression of EED in tumors promotes EMT and is also required for cell proliferation	Lymphoma, Breast cancer, Colorectal cancer	[17,23,200,201,202,203]
			LOF mutation (I363M) in EED reduces PRC2 catalytic activity and causes increased susceptibility to myeloid cancers	Myeloid cancers	[204,205]
		Tumor suppressor	Recurrent inactivation of EED or SUZ12 is found in malignant peripheral nerve sheath tumors (MPNST)	MPNST	[23]
	SUZ12	Oncogenic	SUZ12 is overexpressed in a variety of tumors and exerts oncogenic functions by repressing tumor suppressor genes and promoting oncogene expression	Ovarian cancer, Colorectal cancer, HNSCC, Breast cancer	[201,206,207,208]
		Tumor suppressor	SUZ12 inhibits tumor migration, invasion, and development	Liver cancer, Gliomas, Melanomas	[209,210]
			Inactivation of SUZ12 is found in MPNST and T-cell acute lymphoblastic leukemia (T-ALL)	MPNST, T-ALL	[23,192,211,212]
PRC2.1 or PRC2.2	RBAP46 (RBBP7)	Oncogenic	RBAP46 is required for tumorigenesis	Bladder cancer, Prostate cancer	[213,214]
PRC2.1	EPOP	Oncogenic	The oncogenic role may be mediated through its interaction with EloB/C and USP7 to modulate the chromatin landscape	Colon cancer, Breast cancer	[75]
	PCL1 (PHF1)	Oncogenic	Might contribute to oncogene expression by fusions in chromosomal translocations that alter chromatin accessibility	Ossifying fibromyxoid tumor, Endometrial stromal tumor	[215,216,217,218]
	PCL2 (MTF2)	Oncogenic	Upregulates the expression levels of EED and EZH2 and increases the catalytic activity of PRC2	Gliomas	[219]
		Tumor suppressor	Overexpression of PCL2 stabilizes p53 to promote cellular quiescence	Myeloid leukemia	[220,221]
	PCL3 (PHF19)	Oncogenic	Increases PRC2 activity	Multiple myeloma, Hepatocellular carcinoma, Glioblastoma	[222,223,224]
		Tumor suppressor	Inhibits angiogenesis and invasion of tumor cells	Prostate cancer, Melanoma	[225,226]
PRC2.2	JARID2	Oncogenic	Overexpression of JARID2 promotes invasion and metastasis	Ovarian cancer, Glioma Rhabdomyosarcoma	[227,228,229]
		Tumor suppressor	Inhibition of self-renewal pathways	Myeloid neoplasms	[230]
	AEBP2	Oncogenic	Inactivation of AEBP2 inhibits proliferation and reduces chemoresistance in ovarian cancer cells	Ovarian cancer	[231]

Complex ^a^, different variants of PRC2; Subunit ^b^, defined as a Polycomb group (PcG) protein present in the specific variant; Superscript numbers, references.

**Table 3 ijms-23-09574-t003:** PcG subunit inhibitors undergoing clinical trials.

Target	Drug	Cancer	Phase	NCT ID
EZH2	Tazemetostat (EPZ-6438, E7438) CAS: 1403254-99-8	Follicular lymphoma	3	NCT04224493
		Advanced epithelioid sarcoma	3	NCT04204941
		Follicular Lymphoma	2	NCT05152459
		Diffuse large B-cell lymphoma, Hematologic neoplasms, Mantle cell lymphoma	2	NCT05205252
		Nasal Cancer	2	NCT05151588
		Synovial sarcoma, Epithelioid sarcoma	2	NCT02601950
		Malignant peripheral nerve sheath tumors	2	NCT04917042
		Metastatic melanoma	2	NCT04557956
		Small cell lung cancer (SCLC)	1	NCT05353439
		Metastatic prostate cancer	1	NCT04846478
		Advanced solid tumors, Hematologic tumors	1	NCT04537715
		B-cell non-Hodgkin lymphoma	1	NCT03009344
	Lirametostat (CPI-1205) CAS: 1621862-70-1	Metastatic castration-resistant prostate cancer	2	NCT03480646
		B-cell lymphoma	1	NCT02395601
		Advanced solid tumors	1	NCT03525795
	Valemetostat (DS-3201b) CAS: 1809336-39-7	T-cell lymphoma	2	NCT04703192
		B-cell lymphoma	2	NCT04842877
		SCLC	2	NCT03879798
		Lymphomas	1	NCT02732275
		Metastatic prostate cancer, Urothelial carcinoma, Renal cell carcinoma	1	NCT04388852
	SHR-2554	Advanced solid tumors, B-cell lymphomas	2	NCT04407741
		Advanced breast cancer	2	NCT04355858
		Lymphomas	1	NCT03603951
		Lymphomas	1	NCT05049083
	CPI-0209	Solid tumors	2	NCT04104776
	PF-06821497 CAS: 1844849-11-1	Castration-resistant prostate cancer, Follicular lymphoma, SCLC	1	NCT03460977
EED	MAK-683 CAS: 1951408-58-4	Diffuse large B-cell lymphoma	2	NCT02900651

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
