# Peer review of "Critical Roles of Polycomb Repressive Complexes in Transcription and Cancer"

_ijms, 2022, doi:10.3390/ijms23179574_

Round 1
Reviewer 1 Report
This is a very comprehensive and well-written review of an important research area. The figures and tables provided contain a lot of detailed information that helps to make a complex story more accessible and help with understanding the information provided in the accompanying text.
A few suggestions for further improvements/clarifications:
Abstract: Although redundant, I suggest that the full names of both PRC1 and PRC2 are written out: "Polycomb repressive complex (PRC1), a ubiquitin ligase, and Polycomb repressive complex 2 (PRC2), a histone methyltransferase"
Page 2: "In this review, we provide an overview of the diverse ..."
Page 3: In section 2.1 at the end of the first paragraph there is a reference just labelled as "REF"
In the next paragraph, it is mentioned that PRC1 catalyzes the formation of HY2AK119ub1 - I suggest following this statement with a few sentences explaining what the structural/functional consequences of this modification is thought to be
In section 2.2., it is not clear to me what is meant by "maintain the conformational equilibrium of RNA polymerase II, and inhibit its transcriptional elongation" - are there more details how this mechanism works, specifically how it inhibits elongation?
Page 4: Towards the end of the page is a sentence starting "In which CBX2 and Psc share ..." - this is grammatically not very elegant and should be reformulated.
Page 6: Another poor formulation in a sentence starting with "Another research has subsequently uncovered ...) - suggest changing this to "Further research subsequently uncovered ..."
Page 9: The autocorrection has changed the "C" in "c-MYC" to a capital letter. Even at the beginnign of the sentence, "c-MYC" should have the "c" written as a small letter because it is a specific name.
Author Response
Reviewer 1
Thank you very much for the positive opinion about our studies and the helpful comments. Based on your constructive comments, we have made the corresponding revisions on the manuscript and the following point-by-point responses to the comments. Enclosed please find our details answers.
- Abstract: Although redundant, I suggest that the full names of both PRC1 and PRC2 are written out: "Polycomb repressive complex (PRC1), a ubiquitin ligase, and Polycomb repressive complex 2(PRC2), a histone methyltransferase"
Re: We have revised this according to your helpful comments. (Lines 21-22)
- Page 2: "In this review, we provide an overview of the diverse ..."
Re: We have revised this sentence as you suggested. (Introduction, Line 83)
- Page 3: In section 2.1 at the end of the first paragraph there is a reference just labelled as "REF"
Re: We have added missing references. An in-depth discussion of the specific classification of PRC1 is detailed in REF 5. (Section 2.1, Line 135)
- In the next paragraph, it is mentioned that PRC1 catalyzes the formation of HY2AK119ub1 - I suggest following this statement with a few sentences explaining what the structural/functional consequences of this modification is thought to be.
Re: Based on your constructive comments, we have revised this sentence as follows. A key function of the mammalian PRC1 is to catalyze H2AK119ub1, an epigenetic modification closely associated with PRCs-mediated gene silencing1. (Section 2.1, Lines 136-137)
- In section 2.2., it is not clear to me what is meant by "maintain the conformational equilibrium of RNA polymerase II, and inhibit its transcriptional elongation" - are there more details how this mechanism works, specifically how it inhibits elongation?
Re: Based on your helpful comments, we have added specific details as follows. Moreover, the occupancy of H2AK119ub1 on chromatin is able to maintain the conformational equilibrium of RNA polymerase II (RNAP) and inhibits its facilitating effect on transcription elongation, which supports the gene transcription repression function of PRC12, 3. Phosphorylation of amino acid residues within the carboxy-terminal domain (CTD) of RNAP is associated with transcription initiation, elongation, and termination, where active transcription sites are typically characterized by phosphorylation of its Ser2 residues, whereas inactive or stable genes bind Ser5-phosphorylated RNAP in promoter-proximal regions4. H2AK119ub1 has been reported to inhibit the transcriptional activation of RNAP by balancing the recruitment of Ser5-phosphorylated RNAP and Ser2-phosphorylated RNAP at gene loci2. On the other hand, it can block the binding of RNAP at early stages of elongation by preventing the recruitment of Facilitates Chromatin Transcription (FACT) to the promoter region of transcription3. (Section 2.2, Lines 153-166)
- Page 4: Towards the end of the page is a sentence starting "In which CBX2 and Psc share ..." - this is grammatically not very elegant and should be reformulated. Page 6: Another poor formulation in a sentence starting with "Another research has subsequently uncovered ...) - suggest changing this to "Further research subsequently uncovered ..." Page 9: The autocorrection has changed the "C" in "c-MYC" to a capital letter. Even at the beginnign of the sentence, "c-MYC" should have the "c" written as a small letter because it is a specific name.
Re: Based on your very valuable comments, we have revised these statements. (Lines 201-202, 265-266, 415-416)
Reference
[1] Wang H, Wang L, Erdjument-Bromage H, et al. Role of histone H2A ubiquitination in Polycomb silencing[J]. Nature. 2004, 431(7010):873-878.
[2] Stock J K, Giadrossi S, Casanova M, et al. Ring1-mediated ubiquitination of H2A restrains poised RNA polymerase II at bivalent genes in mouse ES cells[J]. Nature Cell Biology. 2007, 9(12):1428-1435.
[3] Zhou W, Zhu P, Wang J, et al. Histone H2A monoubiquitination represses transcription by inhibiting RNA polymerase II transcriptional elongation[J]. Mol Cell. 2008, 29(1):69-80.
[4] Phatnani H P,Greenleaf A L. Phosphorylation and functions of the RNA polymerase II CTD[J]. Genes Dev. 2006, 20(21):2922-2936.
Reviewer 2 Report
This review describes the molecular action of the Polycomb Group Complex(es), and how its dysregulation is increasingly appreciated as a tumour-driving mechanism. The authors do a good job of describing the composition and roles of the many different PcG complexes, highlight their involvement in different tumour types, and finish by describing the drug development efforts targeting PcG. This is a timely review, and because of the broad interest in epigenetics and cancer should have a wide audience. I have only some minor comments that, if amended may help improve the piece.
1.The review contains many grammatical errors and would benefit from some additional proof-reading.
2. section 2.3. PRC1 in Transcription Activation: While it is certainly possible that PRC1 may function as a transcriptional activator, an alternative explanation is that these effects are indirect i.e., that loss/reduction of PRC1 activity would lead to upregulation of transcriptional regulators that in turn activate target genes. I think it would make sense to point out this alternative model.
3. Page 7, 1st paragraph: It would be important in this section to point out the differences in the profiles of H3K27me1 vs me2 vs me3, since these marks have rather different profiles in the genome.
4. Section 3.2. PRC2 in Transcription Regulation: In the developmental biology field there appears to be a clear distinction between two types of H3K27me3-marked genes: 1) genes that have H3K27me3 together with H3K4me3 are denoted “bivalent genes” and are OFF but rapidly activatable upon subsequent cues, and 2) genes that have H3K27me3 and H2A119ub1, which are OFF and appear to be more difficult to activate at subsequent stages/time-points. These differences in the profile of PRC2 marks could be pointed out.
5. Table 1 and 2: Please add more detail regarding the nature of the genotypes. The text mentioned some point mutations in some PcG genes, but generally is vague regarding the exact mutations, and tables 1 and 2 do not list this information at all.
6. PcG genes can act as either tumour suppressors or oncogenes, depending upon the context. But what is this context? The authors are vague on the molecular genetic nature of the context-dependency of PcG function regarding proliferation. Indeed, this is a poorly understood area. But understanding the context-dependency of PcG activity is critical for the development of safe drug treatment, and it would be pertinent to discuss these issues in a bit more detail.
Author Response
Reviewer 2
Thank you very much for the positive opinion about our studies and the helpful comments. Based on your constructive comments, we have made the corresponding revisions on the manuscript and the following point-by-point responses to the comments. Enclosed please find our details answers.
- The review contains many grammatical errors and would benefit from some additional proof-reading.
Re: The manuscript has been checked thoroughly again and some grammar errors were corrected. To facilitate your review, we have indicated changes against grammatical errors with a green background and underlining, and changes against inappropriate statements with a green background.
- Section 2.3. PRC1 in Transcription Activation: While it is certainly possible that PRC1 may function as a transcriptional activator, an alternative explanation is that these effects are indirect i.e., that loss/reduction of PRC1 activity would lead to upregulation of transcriptional regulators that in turn activate target genes. I think it would make sense to point out this alternative model.
Re: Based on your constructive comments, we have discussed this alternative model as follows. The reduction of PRC1 activity not only inhibits PRC2-mediated gene silencing but also leads to the upregulation of transcriptional regulators that in turn activate target genes. For example, the reduction of H2AK119ub1 occupancy on chromatin disrupts the conformational equilibrium of RNAP, with an increase in Ser2-phosphorylated RNAP and a decrease in Ser5-phosphorylated RNAP, promoting its transcriptional initiation and elongation effects1. Decreased PRC1 activity would also deregulate low-level or inappropriate transcriptional signals from enhancers, thus promoting a transcriptional burst, rather than facilitating gene activation2. (Section 2.3, Lines 272-280)
- Page 7, 1st paragraph: It would be important in this section to point out the differences in the profiles of H3K27me1 vs me2 vs me3, since these marks have rather different profiles in the genome.
Re: We have discussed differences in H3K27me1 vs me2 vs me3 following your valuable comments. Methylation of H3K27 is progressive (H3K27me3 is the result of mono-methylation of H3K27me2) and H3K27me3 is a stable mark3. In contrast to H3K27me3, the importance of H3K27me2 in maintaining gene repression appears limited4. But H3K27me2 is an important intermediary PRC2 product that not only constitutes a substrate for subsequent H3K27me3 formation but may also prevent H3K27 from being acetylated. Acetylated H3K27 is thought to be antagonistic to PcG-mediated gene silencing and is enriched in the absence of PRC25. Unlike H3K27me2/3, H3K27me1 is still detectable in cells carrying non-functional PRC2 and its enrichment correlates with actively transcribed genes6. Exactly how H3K27me1 is generated is still an issue of debate. H3K27me1 may be catalyzed by PRC2, while its presence in actively transcribed genes also results from demethylation of H3K27me2/3 by lysine demethylase 6B (KDM6B) or (UTX histone demethylase) UTX7. (Section 3.1, Lines 302-314)
- Section 3.2. PRC2 in Transcription Regulation: In the developmental biology field there appears to be a clear distinction between two types of H3K27me3-marked genes: 1) genes that have H3K27me3 together with H3K4me3 are denoted “bivalent genes” and are OFF but rapidly activatable upon subsequent cues, and 2) genes that have H3K27me3 and H2A119ub1, which are OFF and appear to be more difficult to activate at subsequent stages/time-points. These differences in the profile of PRC2 marks could be pointed out.
Re: Based on your helpful comments, we have complemented the distinction of the two types of H3K27me3-marked genes as follows. PRC2-catalyzed H3K27me3 is generally considered a hallmark of gene silencing. In the field of developmental biology, there are mainly two types of H3K27me3-marked genes: (1) genes with both H3K27me3 and H3K4me3 are denoted “bivalent genes”, which are OFF but can be rapidly activated under subsequent cues8; (2) genes that have H3K27me3 and H2AK119ub1, which are OFF and appear to be more difficult to activate at subsequent stages. (Section 3.2, Lines 336-341)
- Table 1 and 2: Please add more detail regarding the nature of the genotypes. The text mentioned some point mutations in some PcG genes, but generally is vague regarding the exact mutations, and tables 1 and 2 do not list this information at all.
Re: Following your helpful comments, we have supplemented the specific information of these mutations in tables or in the main text.
Later studies on ESCs with loss-of-function mutations in mouse RING1B (I53A) identified chromatin decompaction, and addition of RING1B restored chromatin compaction in vivo, providing further evidence of a chromatin compaction role for PRC1. (Section 2.2, Lines 192-195)
Missense mutations in PHC3 promote cell proliferation (G201C) (Table 1)
EZH2 is involved in cancer initiation and progression mainly due to the transcriptional repression activity it exerts in the PRC2 complex, and gain-of-function (GOF) mutants of EZH2 (Y647F/N, A677G, and A687V) are frequently generated in several lymphomas, further promoting tumorigenesis. (Section 4.3, Lines 537-540)
LOF mutants of EZH2 (G266E, T393M, and C606Y) that result in loss of PRC2 function and drive Notch signaling activation, and increased the in vivo tumorigenic potential of T-ALL cells, suggesting that PRC2 may have a tumor suppressor function, although the specific mechanisms remain to be further explored. (Section 4.4, Lines 582-586)
For example, in NSCLC with loss-of-function mutations in BRG1 (W764R) or gain-of-function mutations in EGFR (T790M and L858R), inhibition of EZH2 catalytic activity promotes apoptosis and sensitivity to topoisomerase II (TopoII) inhibitors.(Section 4.4, Lines 616-619)
LOF mutation (I363M) in EED reduce PRC2 catalytic activity and cause increased susceptibility to myeloid cancers. (Table 2)
- PcG genes can act as either tumour suppressors or oncogenes, depending upon the context. But what is this context? The authors are vague on the molecular genetic nature of the context-dependency of PcG function regarding proliferation. Indeed, this is a poorly understood area. But understanding the context-dependency of PcG activity is critical for the development of safe drug treatment, and it would be pertinent to discuss these issues in a bit more detail.
Re: Following your helpful comments, we have discussed the context-dependence of PcG activity. Both enhancement and attenuation of PRC2 catalytic activity can promote tumor development, suggesting that PcG genes are context-dependent tumor suppressors or oncogenes. Further observations of the context-dependent roles of PRC2 have revealed that the effects of loss-of-function and gain-of-function alterations do not simply segregate based on tissue or tumor type9. Furthermore, several studies have shown that the role of PRC2 in cancer depends on tumorigenic alterations in other genes10, 11. For example, in NSCLC with loss-of-function mutations in BRG1 (W764R) or gain-of-function mutations in EGFR (T790M and L858R), inhibition of EZH2 catalytic activity promotes apoptosis and sensitivity to topoisomerase II (TopoII) inhibitors. Conversely, in BRG1 wild-type tumors, inhibition of EZH2 upregulates BRG1 and eventually confers stronger resistance to TopoII inhibitors10. Furthermore, in a Kras-driven mouse model of NSCLC, EED loss accelerated or delayed tumor formation depending on p53. In a WT-p53 background, EED loss promotes inflammation, whereas p53 inactivation leads to invasive mucinous adenocarcinoma11. Thus, the context-dependent role of PRC2 suggests that its function in specific cancer types is enormously complex and future work will be beneficial to exhaustively characterize its molecular implications in different cancers that will also help in identifying appropriate approaches to reverse their deregulation in different cells and provide suitable therapies for PRC2-dependent cancers. (Section 4.4, Lines 610-630)
Reference
[1] Stock J K, Giadrossi S, Casanova M, et al. Ring1-mediated ubiquitination of H2A restrains poised RNA polymerase II at bivalent genes in mouse ES cells[J]. Nature Cell Biology. 2007, 9(12):1428-1435.
[2] Dobrinic P, Szczurek A T,Klose R J. PRC1 drives Polycomb-mediated gene repression by controlling transcription initiation and burst frequency[J]. Nat Struct Mol Biol. 2021, 28(10):811-824.
[3] Zee B M, Levin R S, Xu B, et al. In vivo residue-specific histone methylation dynamics[J]. J Biol Chem. 2010, 285(5):3341-3350.
[4] Sarma K, Margueron R, Ivanov A, et al. Ezh2 requires PHF1 to efficiently catalyze H3 lysine 27 trimethylation in vivo[J]. Mol Cell Biol. 2008, 28(8):2718-2731.
[5] Tie F, Banerjee R, Stratton C A, et al. CBP-mediated acetylation of histone H3 lysine 27 antagonizes Drosophila Polycomb silencing[J]. Development. 2009, 136(18):3131-3141.
[6] Cui K, Zang C, Roh T Y, et al. Chromatin signatures in multipotent human hematopoietic stem cells indicate the fate of bivalent genes during differentiation[J]. Cell Stem Cell. 2009, 4(1):80-93.
[7] Swigut T,Wysocka J. H3K27 demethylases, at long last[J]. Cell. 2007, 131(1):29-32.
[8] Blanco E, Gonzalez-Ramirez M, Alcaine-Colet A, et al. The Bivalent Genome: Characterization, Structure, and Regulation[J]. Trends Genet. 2020, 36(2):118-131.
[9] Comet I, Riising E M, Leblanc B, et al. Maintaining cell identity: PRC2-mediated regulation of transcription and cancer[J]. Nat Rev Cancer. 2016, 16(12):803-810.
[10] Fillmore C M, Xu C, Desai P T, et al. EZH2 inhibition sensitizes BRG1 and EGFR mutant lung tumours to TopoII inhibitors[J]. Nature. 2015, 520(7546):239-242.
[11] Serresi M, Gargiulo G, Proost N, et al. Polycomb Repressive Complex 2 Is a Barrier to KRAS-Driven Inflammation and Epithelial-Mesenchymal Transition in Non-Small-Cell Lung Cancer[J]. Cancer Cell. 2016, 29(1):17-31.
Reviewer 3 Report
The manuscript entitled "Critical Roles of Polycomb Repressive Complexes in Transcription and Cancer" discusses recent research in the transcriptional regulation of PRCs, the oncogenic and tumour suppressor roles of PcG proteins and the research progress of inhibitors targeting PRCs. The manuscript is written well and is of prime importance to the scientific community. Some of my observations are:
1. I have searched for similar articles on google and found similar articles already published. Authors must justify novelty in their manuscript over the previously published reviews on the same topic.
2. Include the chemical structures of the inhibitors targeting PRCs discussed in the manuscript (Table 3).
3. Discuss outcomes of clinical trials (Table 3).
Author Response
Reviewer 3
Thank you very much for the positive opinion about our studies and the helpful comments. Based on your constructive comments, we have made the corresponding revisions on the manuscript and the following point-by-point responses to the comments. Enclosed please find our details answers.
- I have searched for similar articles on google and found similar articles already published. Authors must justify novelty in their manuscript over the previously published reviews on the same topic.
Re: First, the focus of our review differs from similar published articles, for variants in PRC1 and PRC2 we do a brief discussion and we focus more on discussing the roles of Polycomb repressive complex complexes (PRCs) in transcription and cancer and the associated mechanisms. Second, in discussing the functions of PRCs in transcription, not only their canonical functions are summarized, but non-canonical functions are also described. Then, in reviewing the dual roles of oncogenic and tumor-suppressor PRCs in cancers, we mainly discuss targeting the constituent subunits of PRCs, which may provide new potential targets for drug development. In addition, we summarize the progress, ideas, clinical experimental results and existing problems of related inhibitors so far, which will provide a certain reference for readers interested in doing the research and development of related inhibitors. Finally, our article tends to express our insights more with figures and tables, which is more beneficial for readers to understand this relatively complex topic of PRCs.
- Include the chemical structures of the inhibitors targeting PRCs discussed in the manuscript (Table 3).
Re: Based on your very valuable comments, we have drawn the chemical structures of inhibitors in the main text and table 3.
Figure 6. Inhibitor structures targeting PRC1 or PRC2.
- Discuss outcomes of clinical trials (Table 3).
Re: Following your constructive comments, we have discussed the results of the clinical trials in table 3. There are also many potent and better bioavailable EZH2 catalytic inhibitors currently undergoing phase 1/2/3 clinical trials, alone or in combination with other drugs, for the treatment of several solid tumors, mainly lymphoma, prostate cancer and, small cell lung cancer (Table 3). In addition, Tazemetostat is being used in several phase 2 clinical trials for the treatment of diffuse large B-cell lymphoma, hematologic neoplasms, mantle cell lymphoma, and peripheral nerve sheath tumors (NCT05205252, NCT04917042). Constellation pharmaceuticals has also tested multiple EZH2 inhibitors in clinical trials, including CPI-1205 and CPI-1205; CPI-1205 (NCT03480646), as their first-generation EZH2 inhibitor, is currently in phase 2 testing for metastatic castration-resistant prostate cancer, while a second-generation EZH2 inhibitor (CPI-0209) is also being tested in phase 2 for solid tumors (NCT04104776). An EZH2 inhibitor (PF-06821497) developed by Pfizer has entered phase 1 testing in patients with small cell lung cancer, castration-resistant prostate cancer and, follicular lymphoma (NCT03460977). (Section 4.5, Lines 704-718)
Round 2
Reviewer 3 Report
The authors successfully amended the manuscript. Now, this manuscript can be accepted for publication.